# Photocatalytic Degradation of Ethiofencarb by a Visible Light-Driven SnIn_4_S_8_ Photocatalyst

**DOI:** 10.3390/nano11051325

**Published:** 2021-05-18

**Authors:** Chiing-Chang Chen, Janah Shaya, Kyriaki Polychronopoulou, Vladimir B. Golovko, Siriluck Tesana, Syuan-Yun Wang, Chung-Shin Lu

**Affiliations:** 1Department of Science Education and Application, National Taichung University of Education, Taichung 403, Taiwan; ccchen@mail.ntcu.edu.tw; 2College of Medicine and Health Sciences, Khalifa University, Abu Dhabi P.O. Box 127788, United Arab Emirates; shaya.janah@ku.ac.ae; 3College of Arts and Sciences, Khalifa University, Abu Dhabi P.O. Box 127788, United Arab Emirates; 4Center for Catalysis and Separation, Khalifa University of Science and Technology, Abu Dhabi P.O. Box 127788, United Arab Emirates; kyriaki.polychrono@ku.ac.ae; 5Department of Mechanical Engineering, Khalifa University of Science and Technology, Abu Dhabi P.O. Box 127788, United Arab Emirates; 6School of Physical and Chemical Sciences, The MacDiarmid Institute for Advanced Materials and Nanotechnology, University of Canterbury, Christchurch 8140, New Zealand; vladimir.golovko@canterbury.ac.nz (V.B.G.); ri30033ta@gmail.com (S.T.); 7School of Medical Applied Chemistry, Chung Shan Medical University, Taichung 402, Taiwan; cindy20122224@gmail.com; 8Department of General Education, National Taichung University of Science and Technology, Taichung 403, Taiwan

**Keywords:** photocatalysis, visible light, SnIn_4_S_8_, ethiofencarb, active radicals, degradation mechanism

## Abstract

This work reports the preparation and detailed characterization of stannum indium sulfide (SnIn_4_S_8_) semiconductor photocatalyst for degradation of ethiofencarb (toxic insecticide) under visible-light irradiation. The as-prepared SnIn_4_S_8_ showed catalytic efficiency of 98% in 24 h under optimal operating conditions (pH = 3, catalyst dosage of 0.5 g L^−1^). The photodegradation reaction followed pseudo-first-order kinetics. The major intermediates have been identified using gas chromatography/mass spectrometry. ^•^O_2_^−^ and ^•^OH radicals appeared to be the primary active species in the degradation process as revealed by scavenger and electronic spin resonance studies, while photogenerated holes had a secondary role in this process. A plausible mechanism involving two routes was proposed for ethiofencarb degradation by SnIn_4_S_8_ after identifying the major intermediate species: oxidative cleavage of the CH_2_-S and the amide bonds of the carbamate moiety. Lastly, SnIn_4_S_8_ was found to be efficient, stable, and reusable in treating real water samples in three successive photodegradation experiments. This study demonstrates the prospect of SnIn_4_S_8_ photocatalysis in treatment of natural and contaminated water from extremely toxic organic carbamates as ethiofencarb.

## 1. Introduction

Carbamates represent an important class of pesticides with different efficient biological activities that led to its frequent utilization worldwide as insecticides, fungicides, and molluscicides, among other uses [1,2]. Carbamates can pose serious environmental risks and health problems. A recent study, for instance, reported that carbamates were associated to cognitive and neuropsychological impairment, acting as acetylcholinesterase inhibitors [3,4]. Carbamates are found on the list of precedence issued by the United States Environmental Protection Agency [5]. The major problem with the wide use of carbamates in agriculture is their accumulation in soil ground and surface water, which might last for years since they are very stable and soluble compounds [6].

Among different carbamates, ethiofencarb [(2-ethylthiomethyl-phenyl)-N-methylcarbamate] is a common insecticide applied to control aphids on fruits, vegetables, and corn [7]. Ethiofencarb has been highlighted in numerous scientific papers as extremely toxic to the central nervous system and potentially carcinogenic and mutagenic. This insecticide has been classified as “dangerous for the environment” by the European Union’s Nordic Council of Ministers and as “highly hazardous” by the World Health Organization [8,9,10]. Ethiofencarb and its degradation products were reported to be frequent contaminants in soil, surface and ground water, as well as food products due to environmental spreading of their residues [7]. Consequently, there is a strong demand for finding efficient water treatment methods to remove these pollutants.

Different water treatment methods have been recently proposed such as ‘‘advanced oxidation processes (AOPs)’’ for degradation of organic pollutants [11]. Specifically, photocatalytic degradation processes generally involve the generation of reactive hydroxyl radicals by exposing semiconductor photocatalysts to ultra violet–visible light [12]. Among the most common photocatalysts, TiO_2_ has been shown to be effective in photodegradation of carbamates [13]. Nevertheless, TiO_2_ has low photocatalytic activity under visible-light activation and it absorbs UV radiation with wavelengths less than 387.5 nm, approximately 4% of sunlight only. High energy demand is needed to conserve the degradation efficiency of TiO_2_ in water treatment applications [14]. These factors still hamper the large-scale applications of TiO_2_ photocatalysis [15].

Research has intense focus on compounds with unique optoelectronic and surface properties such as binary and ternary chalcogenides, which exhibit a range of applications from adsorption/catalysis to photovoltaics and solar energy exploitation [16,17]. The general formulae of these compounds are: A_m_B_n_ for binary chalcogenide (A = Al, Ga, In; B = S, Se, Te; m, n = integer); and AB_m_C_n_ for ternary chalcogenide (A = Cu, Ag, Zn, Cd, Sn, etc.; B = Ga, In; C = S, Se, Te; m, n = integer) [18]. Binary chalcogenides are not resistant to photocorrosion by photogenerated holes and show low chemical stability in photocatalytic processes [19]. In contrast, ternary chalcogenides demonstrate high stability, narrow band gap, and strong absorption of visible light, which are highly desirable properties for photocatalytic processes [20]. Among these ternary chalcogenides, AgInS_2_, CdIn_2_S_4_, CaIn_2_S_4_, SnIn_4_S_8_, and ZnIn_2_S_4_ have been reported to be efficient photocatalysts for visible-light-driven H_2_ evolution or degradation of pollutants [21,22,23,24,25].

In particular, stannum indium sulfide (SnIn_4_S_8_) is a ternary chalcogenide semiconductor with a cubic spinel structure that belongs to the Fd3m space symmetry group. Several potential applications have been demonstrated with this chalcogenide such as high-energy batteries and photocatalysis [24,26,27]. SnIn_4_S_8_ possesses a narrow band gap (Eg ~ 2.3 eV), leading to an obvious absorption of visible light and generation of the electron-hole pairs upon excitation in this region. These features reflect the prospects of SnIn_4_S_8_ for visible-light driven photocatalytic applications [28]. For instance, Yan et al. reported the synthesis of SnIn_4_S_8_ microspheres using a solvothermal synthesis and its catalytic activity in the visible-light region, reaching photodegradation efficiency of 95% of azo-dyes in 3.5 h [24]. Wang et al. described the synthesis of flower-like SnIn_4_S_8_ particles and studied their photocatalytic activity for reduction of Cr(VI) in aqueous solutions, the reduction rate was 97% under irradiation for 60 min [26].

Although removal of carbamate pollutants is an intensive research area, to the best of our knowledge, no investigations of the photocatalyic degradation of ethiofencarb have been reported yet. Considering other carbamates, for example thiobencarb, several efficient photcatalytic processes were previously reported such as using TiO_2_ with an efficiency of 99.8%, MoS_2_ achieving 95% removal efficiency, and BiVO_4_ with 97% degradation rate [5,13,29]. This motivated us to present a systematic study of visible-light driven photocatalytic degradation of ethiofencarb using an efficient ternary chalcogenide, SnIn_4_S_8_, and the influence of various parameters on this process. Secondly, this work provides a better understanding of the contribution of major reactive species in this photodegradation process vis scavenger and ESR experiments. Thirdly, the reaction intermediates of ethiofencarb degradation are investigated, providing plausible mechanistic details of the degradation process with the SnIn_4_S_8_/visible light catalytic system. This report lastly assesses the stability and recyclability of the SnIn_4_S_8_ photocatalyst and its potential uses for practical treatment of environmental water samples.

## 2. Materials and Methods

### 2.1. Materials

Ethiofencarb was purchased from Fluka (99.1%, Seelze, Germany) and used to prepare the stock solutions in de-ionized water (10 mg L^−1^) without any further purification. The purity of the organic compound was analyzed by high performance liquid chromatography (HPLC). The solutions were stored in the dark at 4 °C prior to usage. Stannum indium sulfide (SnIn_4_S_8_) was prepared using the following precursors: tin (IV) chloride pentahydrate (98%) and indium (III) chloride tetrahydrate (97%) from Sigma-Aldrich (St. Louis, MO, USA) in addition to L-cysteine (98%) from Alfa Aesar (Haverhill, MA, USA). The other chemicals used in this study were obtained from Merck (Darmstadt, Germany): ammonium acetate (reagent-grade), nitric acid, sodium hydroxide, and methanol (HPLC-grade). De-ionized water was prepared by purification using a Milli-Q water ion-exchange system (Millipore Co., Burlington, MA, USA) with a resistivity = 1.8 × 10^7^ Ω cm.

### 2.2. Preparation and Characterization of SnIn_4_S_8_

The SnIn_4_S_8_ photocatalyst was prepared by dissolving 1 mmol of SnCl_4_·5H_2_O and 4 mmol of InCl_3_·4H_2_O in deionized water (40 mL). Next, 20 mL of L-cysteine (8 mmol) was added slowly to the solution followed by stirring for 30 min. The obtained slurry was transferred afterwards into Teflon-lined stainless steel autoclave and heated for 24 h at 160 °C. The mixture was cooled in ambient temperature. The resulting precipitate was filtered, washed several times with absolute ethanol and distilled water, and dried for 12 h at 60 °C.

X-ray powder diffraction (XRD) was performed using a PHILIPS X’PERT Pro MPD X-ray diffractometer (Almelo, the Netherlands) with Cu Kα radiation (80 mA and 40 kV). Field emission scanning electron microscopy (FE-SEM) was performed using a field-emission microscope (HITACHI S-4800, Tokyo, Japan) (acceleration voltage = 15 kV). The UV–vis diffuse reflectance spectra was recorded by UV–vis spectrophotometer with an integration sphere (Perkin Elmer Lambda 35, Wellesley, MA, USA). An X-ray photoelectron spectroscope (XPS) measurement was carried out with VG Scientific ESCALAB 250 XPS (Waltham, MA, USA). To generate the Al Kα radiation, 15 kV voltage was, and the correction of the positions of peaks was done by C 1s (284.6 eV).

### 2.3. Apparatus and Instruments

The apparatus used in this study for the photocatalytic degradation experiments has previously been described in the literature [30]. The C-75 Chromato-Vue cabinet of UVP provides a wide area of illumination from the 4W visible-light tubes positioned on two sides of the cabinet interior. The use of relatively low intensity visible-light tubes was necessary in order to obtain slower degradation rates and provide favorable conditions for the identification of the intermediates needed to propose a plausible photodegradation mechanism. Ye et al. [31] indicated that the emission spectrum of the F4T5/CW lamp was similar to solar light. The amount of ethiofencarb in the solutions was determined using a Waters ZQ LC system (Milford, MA, USA) with a binary pump, an autosampler, and a photodiode array detector. The identification of the intermediate products of the degradation reaction was done using solid-phase microextraction (SPME) and gas chromatography/mass spectrometry (GC/MS). The fiber-coating divinylbenzene-carboxen-polydimethyl-siloxane (DVB-CAR-PDMS 50/30 μm) and the SPME holder were obtained from Supelco (Bellefonte, PA, USA). Considering GC/MS, a Perkin-Elmer AutoSystem-XL gas chromatograph interfaced to a TurboMass selective mass detector (Waltham, MA, USA) was utilized. The mineralization of ethiofencarb was evaluated by measuring the total organic carbon (TOC) content with a Dohrmann Phoenix 8000 Carbon Analyzer (Mason, OH, USA). The electronic spin resonance (ESR) signals of the active radicals, trapped by 5,5-dimethyl-1-pyrroline-*N*-oxide (DMPO), were recorded on a Bruker EMX A300-10/12 (Billerica, MA, USA) with a microwave bridge (microwave power, 22.8 mW; microwave frequency, 9.85 GHz; modulation frequency, 100 kHz; modulation amplitude, 1 G).

### 2.4. Procedures and Analysis

Experiments with ethiofencarb should be handled with extreme precautions. Several scientific reports highlight the potential danger of this carbamate (carcinogen, mutagen, and toxic to the central nervous system) [8,9,10]. All catalytic experiments involving ethiofencarb must be carried out strictly in the fume hood using high quality goggles, gloves, and facemasks to avoid any exposure to this compound or contaminated air.

The photocatalytic degradation studies were carried out by mixing ethiofencarb solution (10 mg L^−1^) and a specific dosage of the respective photocatalyst. The different initial pH values in the relevant experiments were maintained by the addition of HNO_3_ (0.1 N) or NaOH (0.1 N) solution. The suspension was stirred magnetically for ca. 30 min in the dark to establish the adsorption/desorption equilibrium, prior to the experiments. Two visible lamps (F4T5/CW, Philips Lighting Co., Taipei, Taiwan) were used for visible-light irradiation with a wavelength range of 400–700 nm. The average light intensity was approximately 1420 lux (measured by a digital luxmeter). A 5 mL aliquot was collected at the given irradiation time intervals. The aliquot was centrifuged to remove the SnIn_4_S_8_ photocatalyst and the quantification of the ethiofencarb residual was then carried out using HPLC. Two kinds of eluents were used: 0.025 M ammonium acetate (solvent A) and methanol (solvent B). LC was performed on an Atlantis^TM^ dC_18_ column (250 mm × 4.6 mm i.d., dp = 5 μm). The flow rate of the mobile phase was 1 mL/min. A linear gradient was run as follows, *t* = 0, *A* = 95, *B* = 5; *t* = 20, *A* = 50, *B* = 50; *t* = 35–40, *A* = 10, *B* = 90; and, *t* = 45, *A* = 95, *B* = 5. The elution was monitored at 220 nm.

Identification of the reaction intermediates was done using SPME-GC/MS. The remaining ethiofencarb compound and its intermediates were extracted by direct immersion of the SPME fiber in the sample solution at ambient temperature for 30 min, with a magnetic stirring of 550 ± 10 rpm. The compounds were thermally desorbed from the fiber to the GC injector for 45 min. Separation was performed in a DB-5 capillary column (5% diphenyl/95% dimethyl-siloxane), 60 m, 0.25-mm i.d., and 1.0-μm thick film. A split-splitless injector was used at 250 °C and a split flow of 10 mL/min. The flow of the helium gas carrier was 1.5 mL/min. The oven temperature was set at 60 °C for 1.0 min and gradually increased by 8 °C min^−1^ till 240 °C, which was maintained for 21.5 min. The total run was 45 min. Electron impact mass spectra were obtained at electron energy of 70 eV and monitored from 20 to 350 *m/z*. The ion source temperature was 220 °C and the inlet line temperature was 250 °C.

### 2.5. Procedure for Degradation of Ethiofencarb in Real Samples

The catalytic activity of SnIn_4_S_8_ was investigated in environmental water samples from Taichung Park lake and river samples from the Han River in Taichung city. To filter the collected samples and remove the suspended solids, 0.45 μm membrane was used. The samples were then stored at 4 °C, protected from sunlight until analysis. Ethiofencarb was added to the samples (10 mg L^−1^) immediately before the photodegradation experiments. The catalytic tests were carried out using 100 mL of real water samples and 0.5 g L^−^^1^ SnIn_4_S_8_ at pH = 3. The photocatalytic activity is analyzed as C/C_0_ as a function of irradiation time (*t*). C represents ethiofencarb concentration at a specific time, and C_0_ is its initial concentration. Error bars represent the standard deviations of duplicate runs.

## 3. Results and Discussion

### 3.1. Characterization of SnIn_4_S_8_

The X-ray diffraction pattern of the as-synthesized SnIn_4_S_8_ powder is presented in Figure 1. According to the literature, the peaks were located at 2θ = 14.0°, 18.6°, 23.3°, 27.4°, 28.7°, 33.2°, 41.0°, 43.6°, 47.8°, 50.1°, 55.9°, 59.4°, and 66.7°. These peaks correspond to (111), (202), (220), (311), (222), (400), (422), (333), (440), (531), (533), (444), and (553) crystalline planes of cubic phase of SnIn_4_S_8_ (JCPDS card #42-1305) [32,33]. The prepared materials show high purity since the spectrum does not show any impurity peak. The morphology of the prepared SnIn_4_S_8_ powder is shown in the field-emission SEM image in Figure 2. Network-like microspheres were observed with an average diameter of 1–2 μm. Close observation shows that the microspheres comprise numerous interleaving flakes with an approximate thickness equal to 10 nm.

The main factor in assessment of the photocatalytic activity of a semiconductor is its optical absorption properties, which depend on its band structure [34,35,36]. Thus, the UV–vis diffuse reflectance spectrum of SnIn_4_S_8_ was recorded (Figure 3a). The as-prepared SnIn_4_S_8_ absorbs ultraviolet and visible light with wavelengths shorter than 550 nm. An onset of the absorption edge is observed starting around 550 nm in the visible light region with steep increase of absorption moving into the ultraviolet region. This absorption of visible light is attributed to the band-gap transition of the catalyst [24]. The equation (αhν)*^n^* = *k*(hν − Eg) can be used to estimate the band gap energy. In this equation, α represents the absorption coefficient, hν refers to the photons’ energy, *k* is the Coulomb constant, Eg denotes the absorption band gap energy, and value of *n* is 2 for a direct band gap semiconductor and 1/2 for an indirect one [26]. Figure 3b shows the plots of (αhν)^2^ as a function of hν of the sample. The band gap (Eg) of SnIn_4_S_8_ was calculated to be 2.23 eV, which is suitable for photocatalytic decomposition of organic compounds under visible-light irradiation.

XPS analysis was used to confirm the composition and valence state of elements of the SnIn_4_S_8_ material (Figure 4). Two intense peaks in the Sn 3d region can be observed in Figure 4a at 486.0 eV (Sn 3d_5/2_) and 494.4 eV (Sn 3d_3/2_), which is relevant to Sn^4+^ oxidation state on the surface of the sample [37]. Considering indium in Figure 4b, the characteristic peaks were located at 444.2 eV and 451.8 eV, which are associated to In 3d_5/2_ and In 3d_3/2_, respectively. These XPS peaks confirm the presence of In^3+^ in the synthesized materials [27]. The S 2p spectrum exhibits a strong peak at 160.8 eV, indicating the presence of the S^2^^−^ species in the sample. Peak deconvolution of S 2p spectrum (Figure 4c) has been performed. The binding energies of 160.8 eV and 161.9 eV correspond to S 2p_3/2_ and S 2p_1/2_, respectively. All these results are in agreement with those previously reported for SnIn_4_S_8_ crystal [33].

### 3.2. Photocatalytic Reaction

#### 3.2.1. Adsorption Experiments and Catalyst Dosage

The loading of the catalyst is a key parameter in catalytic applications [38,39]. First, blank adsorption experiments at different pH values in the dark (without exposure to visible light) revealed that the change of initial ethiofencarb concentrations after 12 h of mixing with SnIn_4_S_8_ was less than 6% (Appendix A). Thus, the adsorption of ethiofencarb on SnIn_4_S_8_ is insignificant and can be neglected.

Next, different dosages of the SnIn_4_S_8_ photocatalyst (from 0.1 to 1.0 g L^−1^) were employed to study the photodegradation rate of ethiofencarb. The results of these experiments are presented in Figure 5. In the absence of the catalyst, ethiofencarb degradation was negligible under visible-light irradiation for 24 h, contrary to the presence of SnIn_4_S_8_ photocatalyst. Increasing the SnIn_4_S_8_ dosage resulted in an increase in the photocatalytic degradation rate until the optimal dosage was reached at 0.5 g L^−1^. The increase in the reaction rate can be explained by the increase in the total surface area/available catalytic active sites for ethiofencarb degradation upon increasing the dosage of the photocatalyst. After the optimal dosage, the increase of SnIn_4_S_8_ concentration may lead to decrease the intensity of available incident visible light as a result of more light scattering and less light penetration. This in turn can restrain the positive effect arising from increasing the photocatalyst dosage, and, thus, reduce the overall rate [40]. The optimal catalytic dosage (0.5 g L^−1^ of SnIn_4_S_8_) was, therefore, used in all subsequent experiments.

#### 3.2.2. Effect of Initial pH Value

Initial pH has been reported to exhibit significanteffects in photocatalytic applications [36]. The impact of initial pH on ethiofencarb degradation was studied and presented in Figure 6. The rates of the photodegradation of ethiofencarb were found to be strongly dependent on the pH of the solution. Ninety-eight percent of ethiofencarb was degraded within 24h at pH = 3, achieving the maximum reaction rate at this optimal pH. Increasing the pH value to 3, 5, 7, and 9 gradually decreased the photocatalytic activity of SnIn_4_S_8_. The respective rate constants were found to be 0.1546 h^−1^, 0.0982 h^−1^, 0.0773 h^−1^, and 0.0661 h^−1^. This decrease in photocatalytic efficiency can be attributed to a Nernstian shift of the band edges to more negative values as the pH value increases, which reduces the oxidation potential of the positive holes [41,42]. The slower kinetics can also be due to an increase in the concentration of hydroxyl groups on the surface at a higher pH value, which may decrease the conduction band electrons due to electron–hole recombination [43]. This latter model of faster hole–electron recombination at higher pH (9.5), has been previously reported [44]. Figure 6b presents the regression analyses based on the first-order reaction kinetics for the photodegradation of ethiofencarb. ln C_0_/C = *k*_app_*t* plots were used to calculate the rate constants (*k*_app_), listed in Table 1 in addition to the linear regression coefficients. The determined values show that the appropriate first-order relationship appears to fit well.

Total organic carbon (TOC) was used to evaluate the mineralization ability of the SnIn_4_S_8_ photocatalyst for ethiofencarb degradation. The TOC removal rate was 42.2% after 24 h of the photocatalytic reaction under visible light. As could be seen, ethiofencarb degradation is much higher than TOC removal with SnIn_4_S_8_ photocatalyst. This great difference implies that the products of ethiofencarb oxidation remained generally at the intermediate product stage under the present experimental conditions.

#### 3.2.3. Effects of Anions

Anionic species are very common in natural and industrial wastewater. In this section, the impact of anions on the photodegradation of ethiofencarb using SnIn_4_S_8_ photocatalyst was studied, and 0.05 M solutions of Na_2_CO_3_, NaCl, NaNO_3_, and Na_2_SO_4_ were used separately to study the influence of carbonate, chloride, nitrate, and sulfate ions on the reaction rate using an initial concentration of 10 mg L^−1^ of ethiofencarb with a 0.5 g L^−1^ of SnIn_4_S_8_. Figure 7 shows that ethiofencarb degradation was slightly inhibited by these anions, leading to prolonged reaction times. Similar inhibition effect has been reported before with anionic species such as NaCl, NaNO_3_, and Na_2_SO_4_, where these anions might compete on the active sites of the photocatalyst surface or lead to its deactivation [45,46]. Other reports showed that such anions might act as scavengers for the active species (positive holes (h^+^) and hydroxyl radicals (**^•^**OH)), producing less reactive radicals such as Cl^•^ or ClOH^−**•**^ (Equations (1) and (2)). The generated less active radicals slow down the rate of photocatalytic degradation [5].
Cl^−^ + h^+^ → Cl^•^(1)
Cl^−^ + **^•^**OH → ClOH^−**•**^(2)

#### 3.2.4. Identifiction of Active Species (Scavenger and ESR Studies)

Different types of active species, such as ^•^O_2_^−^, h^+^, and/or **^•^**OH, can be responsible for the photocatalytic degradation of ethiofencarb. These active species were tested using scavenger experiments with isopropanol (IPA) for hydroxyl radicals, ammonium oxalate (AO) for h^+^, and 1,4-benzoquinone (BQ) for ^•^O_2_^−^ species [47]. Ethiofencarb degradation experiments with SnIn_4_S_8_ catalyst under visible-light irradiation in the presence and absence of these scavengers are summarized in Figure 8. In comparison to absence of scavengers, the addition of 1 mM of BQ (1 mM) strongly inhibited the degradation process, reducing the efficiency from 98% to 58% under 24 h-irradiation. This suggests the important role of the ^•^O_2_^−^ species in the reported degradation. Similar results were obtained upon adding 1 mM of IPA, with efficiency decrease to 76% in 24 h, inferring that **^•^**OH radicals effectively contribute to ethiofencarb degradation. In contrast, inhibition effect was marginally noticed upon adding AO, indicating only an assistant role of the holes in this photodegradation.

Electron spin resonance (ESR) spin-trap technique was further applied to identify the active radicals using 5,5-dimethylpyridine-*N*-oxide (DMPO). The resulting ESR spectra using the SnIn_4_S_8_/visible-light system are presented in Figure 9. In accordance with the scavenger results, the presence of ^•^OH and ^•^O_2_^−^ species was confirmed by the observed characteristic signals for DMPO-^•^OH (intensity ratios of 1:2:2:1) and DMPO-^•^O_2_^−^ (1:1:1:1) spin adducts [48]. Thus, the ethiofencarb degradation with the proposed photocatalytic system is primarily achieved by ^•^O_2_^−^ and ^•^OH radicals, with an assistant contribution from photogenerated holes.

#### 3.2.5. Reaction Pathway of Ethiofencarb Degradation

Since the active radicals that were found to lead this ethiofencarb catalytic degradation are known to be non-selective, many degradation products can be formed. SPME-GC/MS analysis was carried out to identify the intermediate products of this process. Table 2 and Appendix A present the identified intermediates, their retention times, and the mass of the main ions. Using an extensive library search, compound **1** was identified as 2-[(ethylsulfonyl)methyl]phenol with a fit value of 93%. Precisely, the peak at *m*/*z* = 200 observed in its mass spectrum corresponds to the molecular ion [M]^+^ and the peak at *m*/*z* = 107 represents the characteristic ion [C_6_H_4_CH_2_OH]^+^. Compound **2** presented main peaks at *m*/*z* 120 and *m*/*z* 92, corresponding to [C_6_H_4_OCO]^+^ and [C_6_H_4_O]^+^; its molecular mass was 177. It was, thus, identified as 3-methylbenzo[*e*][1,3]oxazine-2,4-dione, based on previous reports of ethiofencarb degradation in other solvents [49,50]. Compound **3** was identified as 3-methylbenzo[*e*][1,3]oxazine-2-one. The compound presented characteristic peaks at *m*/*z* 163 and 106, corresponding to molecular ion and [C_6_H_4_CH_2_O]^+^, respectively. Compound **4** was shown to be 2-[(ethylsulfanyl)methyl]phenol by a library search with a fit value of 92%. The molecular ion peak [M]^+^ appeared at *m*/*z* = 168, and the characteristic ion at *m*/*z* = 107 corresponds to [C_6_H_4_CH_2_OH]^+^ group. Compound **5** showed a peak at *m*/*z* = 166 ([M]^+^) and a characteristic ion at *m*/*z* = 107 ([C_6_H_4_CH_2_OH]^+^), and it was identified as 2-methylbenzo[*e*][1,3]oxathiane, as previously described in ethiofencarb degradation in non-aqueous media [51]. Compound **6** with *m*/*z* = 124 ([M]^+^) and 106 ([C_6_H_4_CH_2_O]^+^) was identified as 2-hydroxybenzyl alcohol. Compound **7** (*m*/*z* = 122 as [M]^+^ and 93 as C_6_H_4_OH]^+^) was identified as 2-hydroxybenzaldehyde by a library search with a fit value of 98%. Compound **8** was a good match of diethyl disulfide (84% fit value). Its retention time was 11.48 min, conforming well to that of pure Et_2_S_2_ in methanol obtained from commercial sources (∼11.4 min). Et_2_S_2_ is an interesting intermediate because it is most likely formed by dimerization of ethanethiol (a molecular segment of ethiofencarb’s side chain). Compound **9** was identified as ethanethiol by a library search with a fit value of 88%. Its molecular ion [M]^+^ was *m*/*z* = 62 and characteristic ions were *m*/*z* = 47 and *m*/*z* = 29, corresponding to [CH_2_=SH]^+^ and [CH_3_CH_2_]^+^, respectively.

A plausible degradation mechanism is depicted in Figure 10 based on the detected intermediate products and the two possible reaction sites on the ethiofencarb molecule. The mechanism consists of two main routes. Route **A** involves oxidative cleavage of the CH_2_-S bond of ethiofencarb proceeded by a cyclization reaction to produce the cyclic adducts **2**–**3**. The photocleavage of the C–S bond has been well reported and is in accordance with its weak bond strength (272 kJ mol^−1^) [52]. Likewise, cyclization routes in ethiofencarb degradation have been documented in the presence of isopropanol and cyclohexane [49]. Route **B** involves the cleavage of the carbamate group, resulting in compound **4** (2-[(ethylsulfanyl)methyl]phenol) and compound **5** (2-methylbenzo[e][1,3]oxathiane). Compound **4** can be further photooxidized to compound **1** (2-[(ethylsulfonyl)methyl]phenol). Another alternative degradation route of compound **4** is also proposed in Figure 10b. This route is initiated by oxidation of intermediate **4** by the attack of the positive holes and the interfacial transfer of a single electron from the sulfur atom, which generates the 2-[(ethylsulfanyl)methyl]phenol cation radical. Different reports have described the formation of such cation radicals in the degradation of other thiocarbamates [53], organophosphorus pesticides [54], and thioethers [55]. Cleavage of the C–S bond in the generated cation radical produces (2-hydroxyphenyl)methylium ion and C_2_H_5_S**^•^** radical. The formed ion is unstable and can be rapidly hydrolyzed into compound **6** (2-hydroxybenzyl alcohol). Compound **6** can be subsequently oxidized to compound **7** (2-hydroxybenzaldehyde). In turn, the produced C_2_H_5_S**^•^** radical transforms to compound **9** (ethanethiol), upon reacting with an H atom. The H atom can originate from proton reduction by photogenerated electrons (H^+^ + e^−^ → H**^•^**), which was previously observed in degradation of fenitrothion [56]. Lastly, two C_2_H_5_S**^•^** radicals can be dimerized to form compound **8** (diethyl disulfide), which was also observed in photodegradations of other thiocarbamates [57]. Disulfide **8** could also be formed via oxidation of thiol **9** as reported elsewhere [58].

#### 3.2.6. Performance of Recycled Catalyst

The stability and reusability of SnIn_4_S_8_ were assessed using circulating runs of visible-light photodegradation of ethiofencarb. After each run, the used photocatalyst was collected, dried, and reused in the next experiment under the same conditions. As shown in Figure 11, the catalytic activity was almost completely retained in three successive runs. This favors the use of SnIn_4_S_8_ photocatalysts in long-term degradation applications, since it demonstrates the absence of photocorrosion and the high stability and efficient reusability of SnIn_4_S_8_ materials.

#### 3.2.7. Photocatalytic Process of Real Water Samples

In this last section, real water samples were used to study the performance of SnIn_4_S_8_ in photocatalysis of ethiofencarb under visible-light irradiation for practical applications. Figure 12 presents the comparison of the degradation rates in deionized water and real water samples. The prepared material was active for photocatalysis in real water, and ethiofencarb was gradually degraded with the increase of irradiation time. The degradation rate was slightly reduced, as compared to that in deionized water. This is typical in the presence of anions (Appendix A) or hydroxyl radical scavengers in real water. In conclusion, SnIn_4_S_8_ photocatalysis showed to be effective and prospective for practical treatment of environmental and contaminated water.

## 4. Conclusions

This work reports the synthesis of SnIn_4_S_8_ photocatalyst and its efficient application in degradation of ethiofencarb carbamate, which is a commonly used toxic insecticide. The catalytic degradation was carried out under visible-light irradiation, reaching 98% removal of ethiofencarb in 24 h. The optimal catalytic loading for ethiofencarb removal was found to be 0.5 g L^−^^1^ of SnIn_4_S_8_, and the optimal pH was 3. It was also found that common inorganic anions that generally exist in natural water systems such as Cl^−^ and NO_3_^−^ can slightly slow down the kinetics of this photodegradation process. Furthermore, the mechanism of the photocatalytic degradation of ethiofencarb was investigated. ^•^OH and ^•^O_2_^−^ radicals were found to be the principal active species in this reaction. Nine intermediate products were identified, and plausible transformation routes were proposed involving the cleavage of either the carbamate group or the C–S bond of ethiofencarb. The SnIn_4_S_8_ material was found to be stable under visible-light irradiation for three successive runs. The photocatalyst was efficient for practical removal of ethiofencarb from environmental water samples (lake and river water), demonstrating its prospect in wastewater treatment.

## Figures and Tables

**Figure 1 nanomaterials-11-01325-f001:**
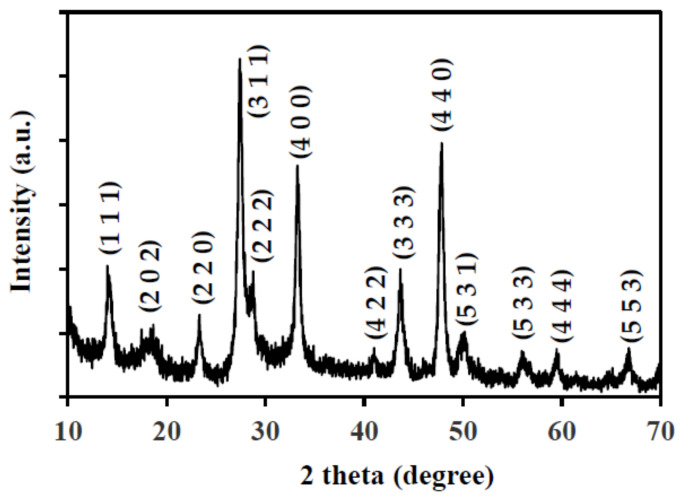
XRD pattern of the as-prepared SnIn_4_S_8_ photocatalyst.

**Figure 2 nanomaterials-11-01325-f002:**
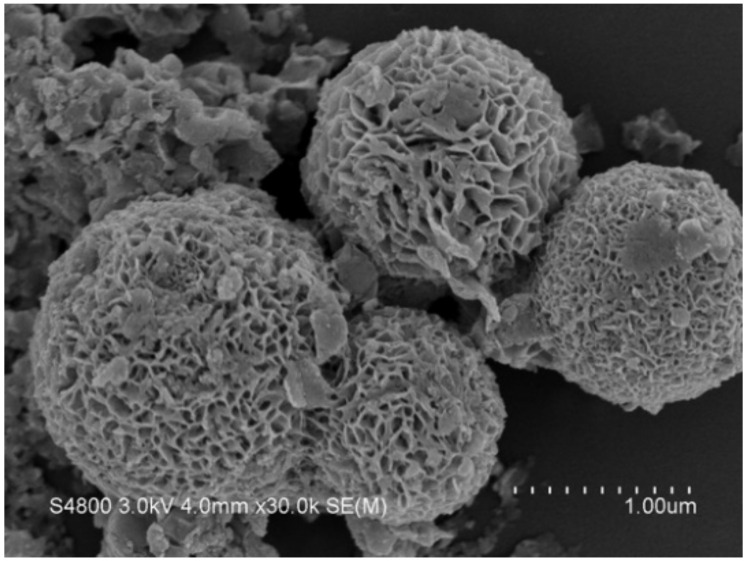
SEM image of the as-prepared SnIn_4_S_8_ photocatalyst.

**Figure 3 nanomaterials-11-01325-f003:**
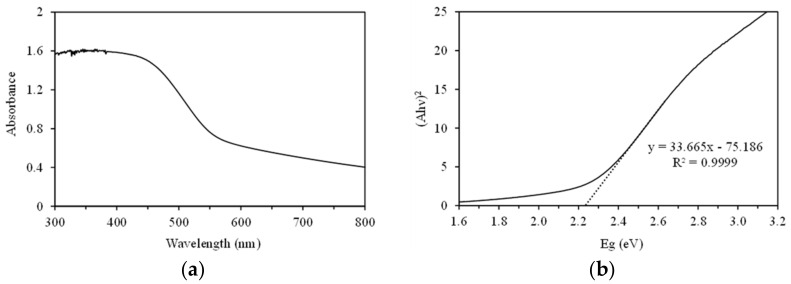
(**a**) UV–vis diffuse reflectance spectrum of the as-prepared SnIn_4_S_8_ photocatalyst; (**b**) Plots of (αhν)^2^ versus energy (hν) for calculating the band gap energy.

**Figure 4 nanomaterials-11-01325-f004:**
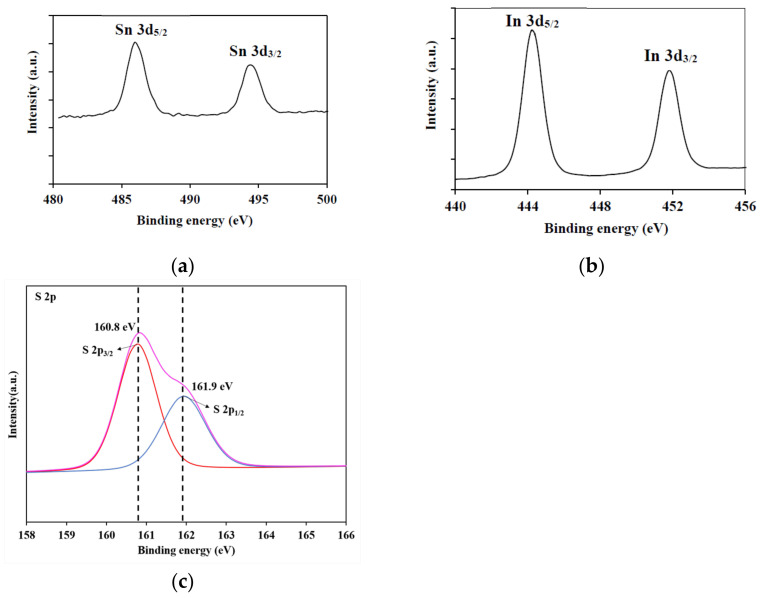
XPS spectra of the as-prepared SnIn_4_S_8_ photocatalyst: (**a**) Sn 3d; (**b**) In 3d; (**c**) S 2p.

**Figure 5 nanomaterials-11-01325-f005:**
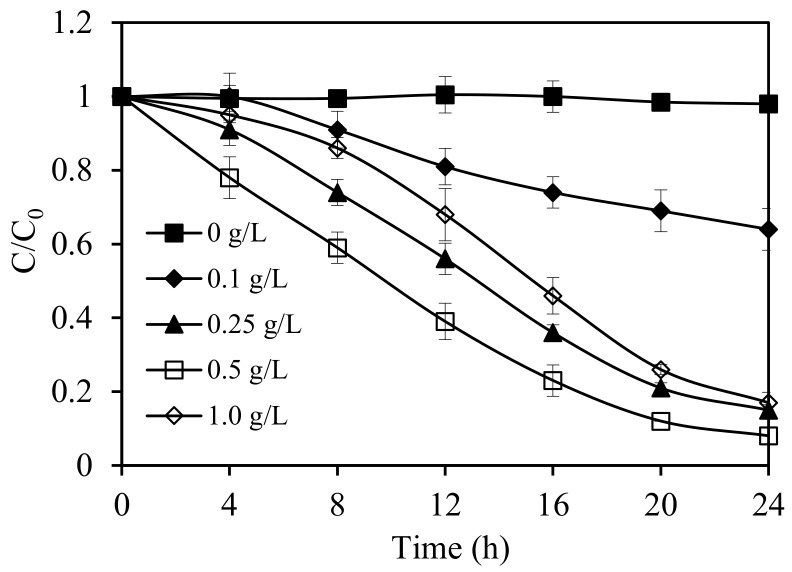
Effect of SnIn_4_S_8_ dosage on the photocatalytic degradation rate of ethiofencarb. Experimental conditions: ethiofencarb concentration 10 mg L^−1^; pH 5.

**Figure 6 nanomaterials-11-01325-f006:**
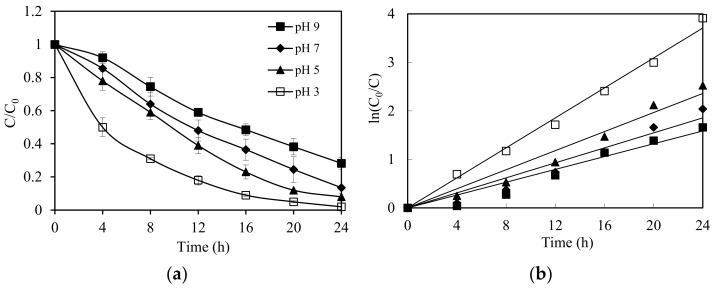
(**a**) pH effect on the photocatalytic degradation rate of ethiofencarb. Experimental conditions: ethiofencarb concentration 10 mg L^−1^; SnIn_4_S_8_ concentration 0.5 g L^−1^. (**b**) Photodegradation kinetics of ethiofencarb in the SnIn_4_S_8_/visible system.

**Figure 7 nanomaterials-11-01325-f007:**
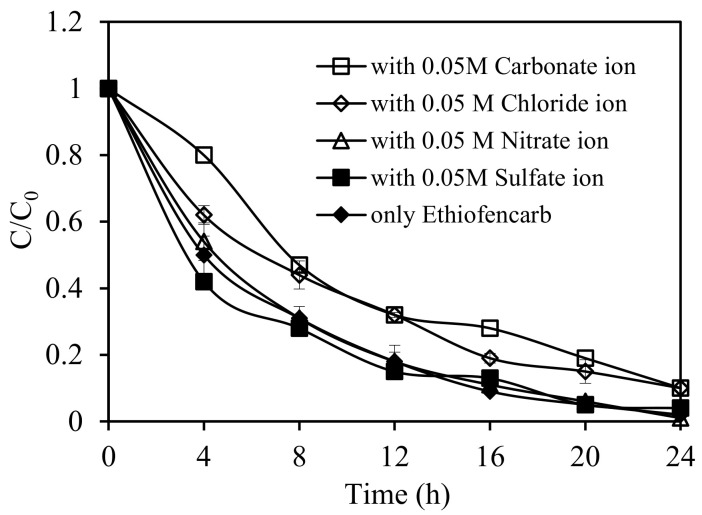
Effect of anions on the photocatalytic degradation rate of ethiofencarb. Experimental conditions: ethiofencarb concentration 10 mg L^−1^; SnIn_4_S_8_ concentration 0.5 g L^−1^; pH 3.

**Figure 8 nanomaterials-11-01325-f008:**
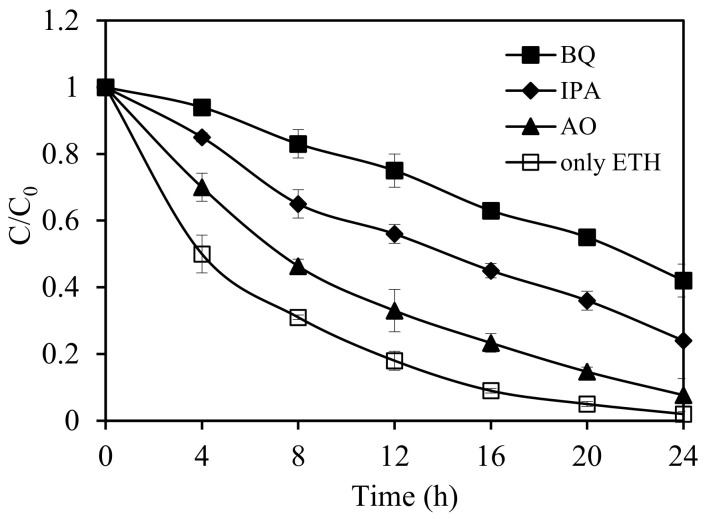
Photocatalytic degradation of ethiofencarb with SnIn_4_S_8_ catalyst in the absence and presence of scavengers (IPA, AO, and BQ) under visible-light irradiation. Experimental conditions: ethiofencarb concentration 10 mg L^−1^; SnIn_4_S_8_ concentration 0.5 g L^−1^; scavenger concentration 1 × 10^−3^ M; pH 3.

**Figure 9 nanomaterials-11-01325-f009:**
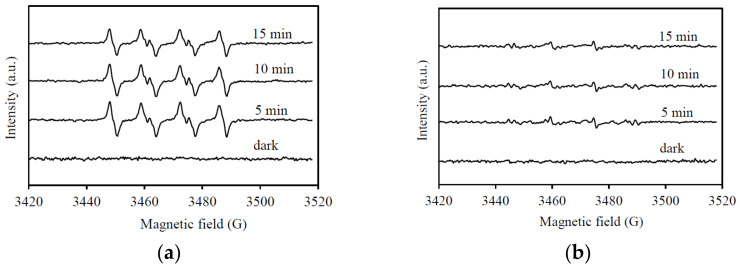
DMPO spin-trapping EPR spectra for (**a**) DMPO-O_2_^−^ and (**b**) DMPO-OH under visible light irradiation with SnIn_4_S_8_ catalyst.

**Figure 10 nanomaterials-11-01325-f010:**
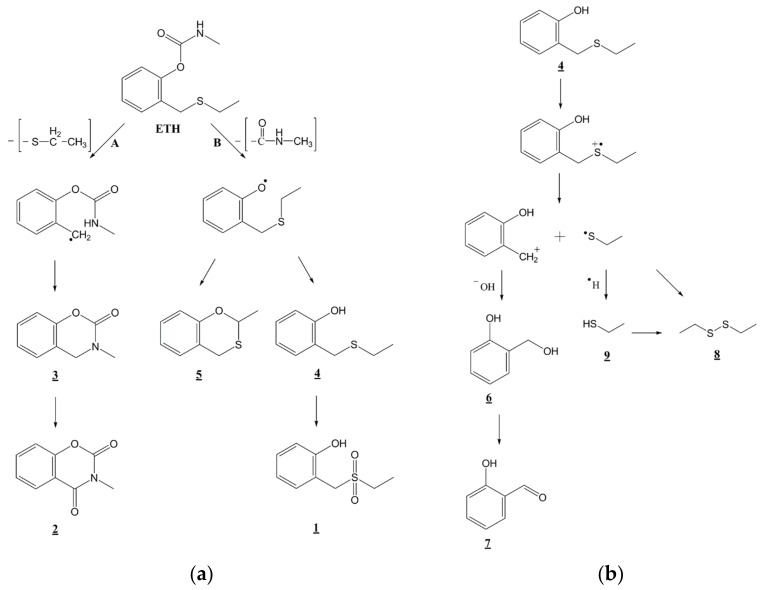
Proposed photocatalytic degradation pathway of (**a**) ethiofencarb and (**b**) 2-[(ethylsulfanyl)methyl]phenol followed by the identification of several intermediates by GC/MS technique.

**Figure 11 nanomaterials-11-01325-f011:**
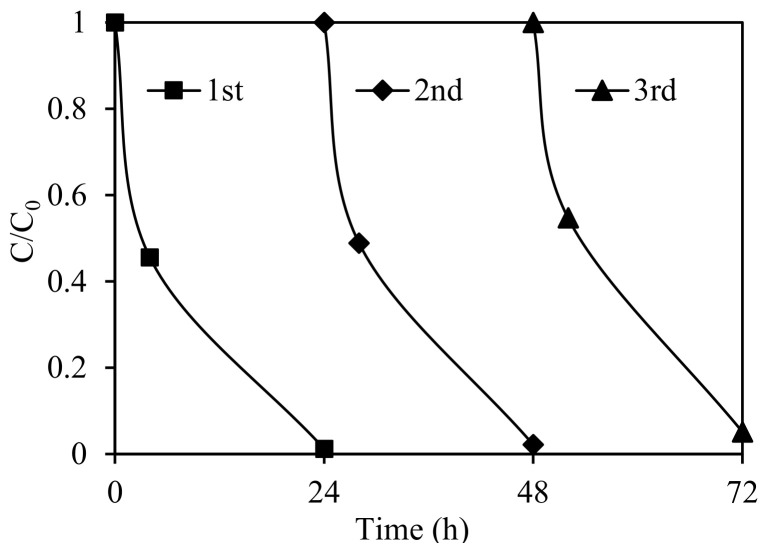
Cycling runs in the photocatalytic degradation of ethiofencarb with SnIn_4_S_8_ catalyst under visible-light irradiation.

**Figure 12 nanomaterials-11-01325-f012:**
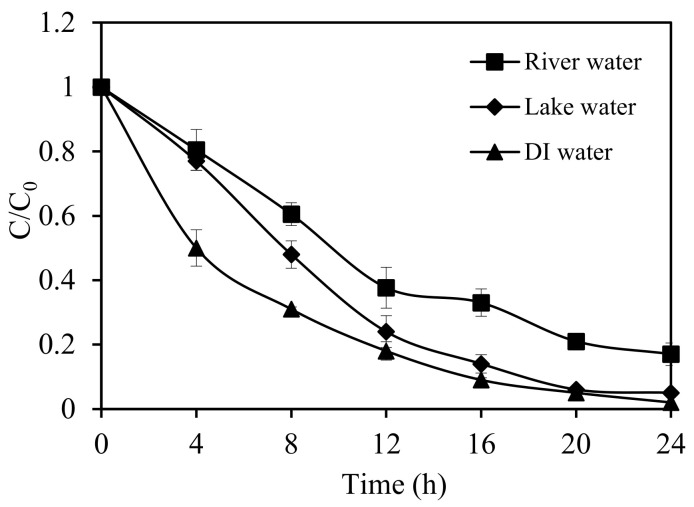
Photocatalytic degradation rates of ethiofencarb in deionized and real water samples. Experimental conditions: SnIn_4_S_8_ concentration, 0.5 g L^−1^; pH 3.

**Table 1 nanomaterials-11-01325-t001:** Photodegradation kinetics parameters (rate constants and linear regression coefficients *R*^2^) for ethiofencarb in the presence of SnIn_4_S_8_ (0.5 g L^−1^).

pH	*k*_app_ (h^−1^)	*R^2^*
3	0.1546	0.998
5	0.0982	0.986
7	0.0773	0.977
9	0.0661	0.978

**Table 2 nanomaterials-11-01325-t002:** Identification of the intermediates from the photodegradation of ethiofencarb by GC/MS.

Peaks	Photodegradation Intermediates	R.T. (min)	MS Peaks (*m*/*z*)
**1**	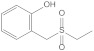	26.36	200, 107, 77
**2**	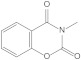	25.07	177, 120, 92
**3**	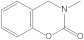	23.88	163, 106, 78
**4**	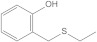	23.74	168, 107, 77
**5**	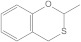	21.08	166, 107, 78
**6**	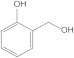	16.56	124, 106, 78
**7**	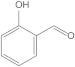	14.28	122, 93, 65
**8**	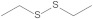	11.48	122, 94, 66
**9**	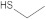	4.27	62, 47, 29

## Data Availability

Data is contained within the article and Appendix A.

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
