# Peer review of "Photocatalytic Degradation of Ethiofencarb by a Visible Light-Driven SnIn4S8 Photocatalyst"

_nanomaterials, 2021, doi:10.3390/nano11051325_

Round 1
Reviewer 1 Report
The manuscript reports results of an experimental study of the photocatlytic degradation of ethiofencarb, a high-toxicity carbamate insecticide, using SnIn4S8, a ternary chalcogenide semiconductor, as photocatalyst, under visible light. The paper claims that SnIn4S8 is a stable catalyst under multiple visible-light irradiation cycles, efficient in practical removal of ethiofencarb from environmental water samples. Moreover, the manuscript claims that plausible photodegradation routes include cleavage of the carbamate group or the C–S bond.
The paper fits the scope of the journal and the research is, to the best of my knowledge, novel. Although the removal of a highly toxic pollutant ethiofencarb from environmental water is important, the impact of the manuscript is difficult to assess, as it does not compare the photodegradation efficiency of the catalyst to the top performers on similar carbamate insecticide compounds. Questions that need to be addressed are i) how does the photodegradation performance of SnIn4S8 on ethiofencarb compare to its efficiency on other pollutants? and ii) how does SnIn4S8 compare to top photocatalysts on similar pollutants?
The state of the art is described, the knowledge gap and the objectives are clear. The methodology seems overall sound but some questions arise regarding the irradiation with visible light. To obtain replicable and, more importantly, comparable results the light source should mimic as well as possible the solar irradiation spectrum. That means that the lamp should reproduce the AM1.5 Global solar irradiation, according to ASTM G-173-03 standards (https://www.nrel.gov/grid/solar-resource/spectra-am1.5.html). The solar irradiance of 1000 W/m2 under AM1.5 conditions corresponds to about 120000 lux (DOI 10.21227/mxr7-p365) illuminance, which is much larger than the 2*1420 lux illuminance provided by the two lamps used in the experiments. Moreover, as the F4T5/CW light sources from Philips Lighting Co. are fluorescent lamps, they may have a significant contribution in the UV region, unlike the solar spectrum, with only about 4%, as rightfully stated in the manuscript.
In conclusion, the manuscript reports interesting new results but the manuscript requires a revision before it can be accepted for publication.
Reviewer 2 Report
This paper describes studies on the preparation and characterization of stannum indium sulfide semiconductor. The obtained sample was studied as photocatalysts for ethiofencarb degradation under Vis light irradiation. While the paper provides some interesting information it could be strengthened further if some additional information could be provided, as outlined below. The manuscript can be published after MINOR revision.
- The abstract is too long. Proper abstract should be a clear, concise summary informative rather than descriptive giving the scope and purpose, methods or procedures, significant new results, and conclusions. It should be corrected.
- Page 4 – “The different initial pH values in the relevant experiments were maintained by the addition of HNO3 or NaOH solution” – concentration of HNO3 and NaOH solution should be added.
- Page 4 - “The suspension was stirred magnetically for ca. 30 min in the dark to establish the adsorption/desorption equilibrium, prior to the experiments” – from the presented data we cannot see that the adsorption/desorption equilibrium was established after 30 min. in the dark. The data should be added. Moreover, the data of adsorption of ethiofencarb on SnIn4S8 for different conditions (pH etc.) should be also presented in the manuscript.
- Figure 1 – the quality of this figure is very low. Please improve.
- Figure 1 – there is a broad peak at 18° in XRD pattern. One should explain the origin of this peak.
- Figure 4 - the quality of this figure is low. Please improve.
- Page 11 - “After each run, the used photocatalyst was properly collected and reutilized in the next experiment” – Was the catalyst regenerated after each cycle? If yes, in what way?
Reviewer 3 Report
Manuscript Number: nanomaterials-1135097
Title: Photocatalytic degradation of ethiofencarb by a visible light-driven SnIn4S8 photocatalyst
The manuscript presents a systematic study related to the obtaining, characterization and application of SnIn4S8 semiconductor material as efficient photocatlyst for toxic ethiofencarb insecticide degradation in visible light irradiation. After the parameters (photocatalyst dosage, pH) of photodegradation process were optimized, the influence of common inorganic anions, which usually are present in natural water systems, and the active radicals involved in the photodegradation of ethiofencarb were investigated. Based on the identification,of reaction intermediates for ethiofencarb degradation, the authors proposed plausible mechanistic details of the degradation process with the SnIn4S8/visible light catalytic system. Moreover, the study pointed out that SnIn4S8 material is a stable photocatalyst under visible-light irradiation for three successive runs, and it is efficient in practical removal of ethiofencarb from real water samples, thus demonstrating its applicability in wastewater treatment.
In general, the paper is well written, but still requires few revisions, especially in terms of form, before publication in Nanomaterials journal.
Here I list the revisions I proposed to be considered by the authors:
In Abstract:
- Line 29: “conformed to” must be replaced with “followed the” or “fits to”
- In Introduction:
- Line 46: “diverse” must be replaced with “different “;
- Line 57: “documented” must be replaced with “highlighted”;
- Lines 76: it is chalcogenides;
- In Materials and Methods:
- Line 125: “was allowed to cool down to” will be replaced with “was cooled in”; “The resulting solid…” what means solid? Precipitate? Solid compound? Solid material???
- In Results and Discussion:
- Line 193: “in line to” must be replaced with “according to”;
- Line 194: “respectively” must be deleted;
- Line 196: “fabricated” must be replaced with “prepared”;
- Line 206: “The major factor of determining the photocatalytic activity of a semiconductor is
- generally its optical..” must be reformulated as “The main factor in the photocatalytic activity assessment of a semiconductor is its…” (suggestion);
- Lines 210-212: please reformulate the text “An intense absorption with a steep edge is observed in the visible light region, which infers that this absorption of visible light originates from the band-gap transition and not from transitions in impurity levels [24]” because is too long and unclear;
- Line 221: “utilized” must be replaced with “used”;
- Line 247: “in all the proceeding experiments” must be replaced with “in all experimental procedures” or “in subsequent experiments”;
- Line 252: “dramatic” must be replaced with “significant”;
- Lines 275-277: The text must be revised as “In this section, we set up to study the impact of anions on the photodegradation of ethiofencarb, using the SnIn4S8 photocatalyst, was studied/investigated”.
- Line 278: “respectively” must be deleted;
- Line 285: “for example” must be deleted;
- Line 302: “reported” must be replaced with “ethiofencarb”;
- Line 314: “In line with” must be replaced with “In accordance with”;
- Line 330: “major” must be replaced with “main”;
- Line 356: “in line with” must be replaced with “in accordance with”;
- In Conclusion:
- Line 407: please specify the pH of the medium, eventually in parentheses;
- Line 413: “The SnIn4S8 material was found to be a stable catalyst under..”
Reviewer 4 Report
The manuscript reports the application of a ternary chalcogenide to the photocatalytic degradation of ethiofencarb. The novelty is not too high and some additional experiments could help to complete and enrich the manuscript.
- Values of Total Organic Carbon (TOC) or equivalent parameter (as COD for example) are essential when application to real water are involved. The toxicity of final effluent is vital to validate the harmless of resulting water discharge.
- S2p peak observed in XPS spectrum, shows a shoulder at lower B.E. Peak deconvolution should be performed and the contribution of each signal assigned.
- The absorption of irradiation is intimately associated to the hydrodynamic particle size of the catalyst in suspension. Did authors measure the aggregation of catalyst particles in the reaction medium? Could the optimal catalyst loading be related to the aggregation induced by high loading? These measurements could enrich the discussion and hold the argumentation.
- The anions effect should be completed with the study of other anions detected in the used real water matrices, at analogous concentration.
- The main characteristics of used real waters should be reported.
Another minor comments:
L293 Check “Identification…”
L300 Check duplicity of “1mM”
L413 Check the sentence “…was found to stable catalyst …”
Reviewer 5 Report
The conclusions seem not clear, bu the difficult part of this ms is that the prcautions taken to protect the operator and the environment are not illustrated.
Round 2
Reviewer 1 Report
The revised manuscript has addressed the issues I raised. The suggestions regarding the light sources used may be difficult to apply to the present study but the authors added a caveat to the text indicating the limitation. They also agreed that the comment was valid and indicated that they would use better light sources in the future. Under these circumstances I think that the paper can be published.
Reviewer 5 Report
in my previous report I listed some comments on the submitted ms, that was meant to increase the palatabilty of the ms for readers. I id not see any any interest for my comments.
